# A novel imaging method (FIM-ID) reveals that myofibrillogenesis plays a major role in the mechanically induced growth of skeletal muscle

Kent W Jorgenson[1], Jamie E Hibbert[1], Ramy KA Sayed[1,2], Anthony N Lange[1], Joshua S Godwin[3], Paulo HC Mesquita[3], Bradley A Ruple[3], Mason C McIntosh[3], Andreas N Kavazis[3], Michael D Roberts[3], Troy A Hornberger[1]*

[1]School of Veterinary Medicine and the Department of Comparative Biosciences, University of Wisconsin-Madison, Madison, United States; [2]Department of Anatomy and Embryology, Faculty of Veterinary Medicine, Sohag University, Sohag, Egypt; [3]School of Kinesiology, Auburn University, Auburn, United States

*For correspondence:
troy.hornberger@wisc.edu

Competing interest: The authors declare that no competing interests exist.

**Abstract** An increase in mechanical loading, such as that which occurs during resistance exercise, induces radial growth of muscle fibers (i.e. an increase in cross-sectional area). Muscle fibers are largely composed of myofibrils, but whether radial growth is mediated by an increase in the size of the myofibrils (i.e. myofibril hypertrophy) and/or the number of myofibrils (i.e. myofibrillogenesis) is not known. Electron microscopy (EM) can provide images with the level of resolution that is needed to address this question, but the acquisition and subsequent analysis of EM images is a time- and cost-intensive process. To overcome this, we developed a novel method for visualizing myofibrils with a standard fluorescence microscope (fluorescence imaging of myofibrils with image deconvolution [FIM-ID]). Images from FIM-ID have a high degree of resolution and contrast, and these properties enabled us to develop pipelines for automated measurements of myofibril size and number. After extensively validating the automated measurements, we used both mouse and human models of increased mechanical loading to discover that the radial growth of muscle fibers is largely mediated by myofibrillogenesis. Collectively, the outcomes of this study offer insight into a fundamentally important topic in the field of muscle growth and provide future investigators with a time- and cost-effective means to study it.

## eLife assessment

The work by Hornberger and team presents a novel workflow for the visualisation of myofibrils with high resolution and contrast that will be highly valued by the scientific community. The methods include **solid** validation of both sample preparation and analysis, and have been used to make the **fundamental** discovery of myofibrillogenesis as the mechanism of mechanical loading-induced growth. Whether this mechanism is present in other settings of muscle growth (i.e., non-loading), other striated tissue (e.g myocardium), or is sex-dependent, will require future experiments.

## Introduction

Comprising ~45% of the body's mass, skeletal muscles are not only the motors that drive locomotion, but they also play a critical role in respiration, whole-body metabolism, and maintaining a high quality of life (*Izumiya et al., 2008*; *Srikanthan and Karlamangla, 2011*; *McLeod et al., 2016*; *Seguin and*

**eLife digest** Approximately 45% of human body mass is made of skeletal muscle. These muscles contract and relax to provide the mechanical forces needed for breathing, moving, keeping warm and performing many other essential processes. Both sedentary and active adults lose approximately 30-40% of this muscle mass by the age of 80, increasing their risk of disease, disability and death. As a result, there is much interest in developing therapies that can restore, maintain and increase muscle mass in older individuals.

Muscles are made of multiple fibers that are in turn largely composed of smaller units known as myofibrils. Previous studies have shown that performing resistance training or other exercise that increases the mechanical loads placed on muscles stimulates muscle growth. This growth is largely due to increased girth of the existing muscle fibers. However, it remained unclear whether this was due to myofibrils growing in size, increasing in number, or a combination of both.

To address this question, Jorgenson et al. developed a fluorescence imaging method called FIM-ID to count the number and measure the size of myofibrils within cross-sections of skeletal muscle. Using FIM-ID to study samples of mouse and human muscle fibers then revealed that increasing mechanical loads on muscles increased the number of myofibrils and this was largely responsible for muscle fiber growth.

FIM-ID mostly relies on common laboratory instruments and free open-source software is used to count and measure the myofibrils. Jorgenson et al. hope that this will allow as many other researchers as possible to use FIM-ID to study myofibrils in the future. A better understanding of how the body controls the number of myofibrils may lead to the development of therapies that can mimic the effects of exercise on muscles to maintain or even increase muscle mass in human patients.

---

*Nelson, 2003*). Indeed, both sedentary and active adults will lose 30–40% of their muscle mass by the age of 80, and this loss in muscle mass is associated with disability, loss of independence, an increased risk of morbidity and mortality, as well as an estimated $40 billion in annual hospitalization costs in the United States alone (*Seguin and Nelson, 2003*; *Janssen et al., 2004*; *Proctor et al., 1998*; *Pahor and Kritchevsky, 1998*; *Goates et al., 2019*). Thus, the development of therapies that can restore, maintain, and/or increase muscle mass is of great clinical and fiscal significance. However, to develop such therapies, we will first need to define the basic mechanisms that regulate skeletal muscle mass.

Skeletal muscle mass can be regulated by a variety of different stimuli with one of the most widely recognized being mechanical signals (*Bodine, 2013*; *Adams and Bamman, 2012*; *Goldberg et al., 1975*; *Roberts et al., 2023*). For instance, a plethora of studies have shown that an increase in mechanical loading, such as that which occurs during resistance exercise (RE), can induce radial growth of the muscle fibers (*Conceição et al., 2018*; *Ema et al., 2016*; *Williams and Goldspink, 1971*; *Haun et al., 2019*; *Jorgenson et al., 2020*). Surprisingly, however, the ultrastructural adaptations that drive this response have not been well defined (*Haun et al., 2019*; *Jorgenson et al., 2020*; *Roberts et al., 2020*). Indeed, a number of seemingly simple and fundamental important questions have not been answered. For instance, during radial growth, the cross-sectional area (CSA) of the muscle fiber increases, but whether this is mediated by an increase in the CSA of the individual myofibrils (i.e. myofibril hypertrophy) and/or the number of myofibrils (i.e. myofibrillogenesis) has not been resolved (*Jorgenson et al., 2020*; *Roberts et al., 2020*; *Wang et al., 1993*; *Ashmore and Summers, 1981*).

A longstanding model in the field proposes that the radial growth of muscle fibers can be explained by a process called 'myofibril splitting' (*Goldspink, 1970*; *Goldspink, 1971*; *Patterson and Goldspink, 1976*; *Goldspink, 1983*). The myofibril splitting model was developed by Goldspink in the 1970s and contends that, during radial growth, myofibrils initially undergo hypertrophy. Then, as the CSA of the myofibrils increases, the outward radial force that they exert on the Z-disc increases. The outward radial forces place a strain on the center of the Z-disc, and when the forces reach a critical threshold the Z-disc will break. It is proposed that the break will begin at the center of the Z-disc and form a longitudinal split that will then propagate through the remainder of the myofibril and, in turn, result in the formation of two smaller daughter myofibrils. It is also proposed that the daughter myofibrils can undergo a cycle of further hypertrophy and splitting, and this cycle will continue until the radial growth of the muscle fiber ceases.

In support of his model, Goldspink published numerous images of single myofibrils that appeared to split into two smaller daughter myofibrils, as well as quantitative data which demonstrated that longitudinal splits usually occur in myofibrils that are significantly larger than myofibrils that do not contain splits (*Goldspink, 1970*; *Goldspink, 1971*; *Patterson and Goldspink, 1976*). Indeed, the model seemed so convincing that, when discussing how an increase in mechanical loading leads to the growth of muscle fibers, Goldspink concluded "*The number of myofibrils increases (Goldspink and Howells, 1974Goldspink and Howells, 1974*), *and this is almost certainly due to the longitudinal splitting of existing myofibrils rather than de novo assembly*" (*Goldspink, 1983*). This statement has been highlighted because, for decades, the myofibril splitting model has served as the textbook explanation of how muscle fibers undergo radial growth (*MacIntosh and McComas, 2006*; *Komi, 2008*; *Bourne, 1972*). However, after performing an exhaustive review of the literature, it was clear that this model has not been rigorously tested (*Jorgenson et al., 2020*). In fact, in many instances, the potential implications of the model are based on erroneous claims. For instance, in Goldspink's aforementioned conclusion, neither reference (*Goldspink and Howells, 1974*), nor any other study that we are aware of, has ever shown that an increase in mechanical loading leads to an increase in the number of myofibrils per fiber. Thus, one is left to wonder why this widely cited model of growth has not been rigorously tested.

When considering currently available technologies, electron microscopy (EM) would be viewed as the gold standard method for testing the myofibril splitting model; however, EM has several drawbacks/limitations. For instance, electron microscopes are highly specialized instruments, and their use requires immense training and potentially cost-prohibitive user fees. Furthermore, the extensive fixation and processing required for EM often yields samples that are not readily amenable to immunolabeling (*D'Amico et al., 2009*), and EM images typically do not have a high degree of contrast between the area that is occupied by the myofibrils and the intermyofibrillar components that surround the myofibrils (e.g. the sarcoplasmic reticulum [SR], see Figure 4A). The latter point is particularly important because it serves as a barrier to the development of programs that can perform automated measurements of features such as myofibril CSA and myofibril number. Thus, with EM, such measurements would need to be manually derived.

To gain insight into the effort that manual measurements of myofibril CSA would require, we collected pilot data from EM images of mouse skeletal muscles. We then used Cochran's formula to determine how many myofibrils per fiber would have to be measured to estimate the mean myofibril CSA with a 5% margin of error at a 95% level of confidence (*Cochran, 1977*). The outcomes revealed that >180 myofibrils per fiber would have to be manually traced. Bear in mind that estimating the average myofibril CSA for the entire sample would require measurements from multiple fibers. For instance, if just 20 fibers per sample were analyzed, it would require the manual tracing of >3600 myofibrils. Taking this a step further, it would mean that a two-group comparison with 7 samples per group would require the manual tracing of >50,000 myofibrils. In our hands, it took 1 hr to measure ≈125 myofibrils, and thus the simple two-group comparison described above would require ≈400 hr of manual tracing. With numbers like this, it was easy to appreciate why the basic elements of Goldspink's model have not been rigorously tested. Thus, the first goal of this study was to develop a time- and cost-effective method for visualizing myofibrils with a level of resolution and contrast that would support automated measurements of myofibril size and myofibril number. With this method in hand, we then set out to answer the fundamentally important question of whether the radial growth of muscle fibers that occurs in response to an increase in mechanical loading is mediated by myofibril hypertrophy and/or myofibrillogenesis.

## Results
### Optimization of the fixation, cryoprotection, and sectioning conditions for FIM-ID

As explained in the Introduction, one of the primary goals of this study was to develop a time- and cost-effective method for visualizing myofibrils with a level of resolution and contrast that would support automated measurements of myofibril size. To accomplish this, we took advantage of the fact that myofibrils are largely surrounded by an SR that is highly enriched with an enzyme called the sarco(endo)plasmic reticulum calcium-ATPase (SERCA) (*Jorgensen et al., 1982*; *Rossi and Dirksen,*

*2006*; *Brandl et al., 1987*). There are two major isoforms of SERCA (SERCA1 which is found in Type II fibers, and SERCA2 which is found in Type I fibers), and monoclonal antibodies for each isoform have been in existence for many years (*Brandl et al., 1987*; *Braun et al., 2022*; *Johansson et al., 2003*; *Majerczak et al., 2008*; *Periasamy and Kalyanasundaram, 2007*). Importantly, these antibodies have been shown to possess excellent reactivity in both flash-frozen and aldehyde-fixed tissues (*Nguyen et al., 2005*; *Rudolf et al., 2006*; *Toral-Ojeda et al., 2016*). Thus, we reasoned that these antibodies could be used to illuminate the periphery of the myofibrils in muscles that had been subjected to a variety of different preservation conditions.

To test our hypothesis, we first performed immunohistochemistry (IHC) for SERCA1 on cross-sections of mouse plantaris muscles that had been flash-frozen in optimal cutting temperature compound (OCT). As shown in *Figure 1A*, small patches within the fibers revealed seemingly intact myofibrils, but the overall integrity of the myofibrils was very poor. This was not surprising because it is known that flash-frozen muscles can suffer from freezing artifacts that present as holes within the fibers (*Meng et al., 2014*; *Nix and Moore, 2020*; *Gaspar et al., 2019*). Thus, to determine if the integrity of the myofibrils was being distorted by freezing artifacts, we co-stained the cross-sections with phalloidin, and the results confirmed that regions of poor myofibril integrity were directly associated with the presence of freezing artifacts (*Figure 1B*). As such, it was concluded that the development of a time- and cost-effective method for visualizing intact myofibrils would require the use of a procedure that eliminates freezing artifacts.

In 1973, Tokuyasu described a fixation and cryoprotection procedure that could preserve the ultrastructure of skeletal muscle at a level that was equivalent to that obtained with the conventional resin-embedding procedures of EM (*Tokuyasu, 1973*). Importantly, unlike resin-embedded tissues, it was shown that tissues preserved with the Tokuyasu method retain a high level of immunoreactivity and are therefore amenable to immunolabeling (*Möbius and Posthuma, 2019*). Indeed, in 1982, the Tokuyasu method was used to define the ultrastructural localization of SERCA (*Jorgensen et al., 1982*). The key components of the Tokuyasu method involve an aldehyde-based fixative and sucrose which not only serves as a cryoprotectant but it also controls the consistency of the tissue to allow for ultrathin (<0.2 µm) sectioning. Specifically, Tokuyasu demonstrated that with a proper combination of the sucrose and sectioning temperature, one could obtain ultrathin sections of various tissues (*Tokuyasu, 1973*). However, the traditional Tokuyasu method requires sectioning temperatures that range from –50°C to –110°C, and such sectioning temperatures are dependent on the use of a cryo-ultramicrotome which is a rare and expensive instrument. Hence, although it appeared that the traditional Tokuyasu method would enable us to obtain immunoreactive sections that were devoid of freeze artifacts, the need for a cryo-ultramicrotome defeated our goal of developing a cost-effective method. Nevertheless, we were inspired by the Tokuyasu method, and we predicted that the right combination of sucrose and sectioning temperature would enable us to obtain adequately cryoprotected and immunoreactive semi-thin sections with a regular cryostat.

To examine the validity of our prediction we performed a lengthy series of trial-and-error experiments, and through these experiments, we discovered that SERCA immunoreactive cross-sections that were devoid of freeze artifacts could be obtained from muscles that had been extensively fixed with 4% paraformaldehyde and subsequently cryoprotected with 30–45% sucrose (*Figure 1C–D*). Qualitatively, we concluded that the best integrity of the myofibrils was found in samples that had been cryoprotected with 45% sucrose and we also discovered that by lowering the sectioning temperature to –30°C we could eliminate sectioning artifacts that were readily present on the surface of samples that had been sectioned at warmer temperatures (*Figure 2*). Thus, whenever possible, all future studies were performed on tissues that had been fixed with 4% paraformaldehyde, cryoprotected with 45% sucrose, and sectioned at –30°C.

As shown in *Figure 1C*, immunolabeling of SERCA1 on muscles that had been subjected to the optimized fixation/cryoprotection/sectioning procedure led to a moderately well-contrasted illumination of the periphery of the myofibrils on a standard widefield fluorescence microscope. We chose to use a widefield fluorescence microscope for image acquisition because the per frame capture time on a widefield fluorescence microscope is extremely short when compared with alternatives such as a confocal fluorescence microscope. For instance, our image acquisition parameters required <1 s per frame on a Leica Widefield Thunder Microscope compared with >3 min per frame on a Leica confocal SP8 microscope. In other words, the use of widefield microscopy aligned with our goal of developing

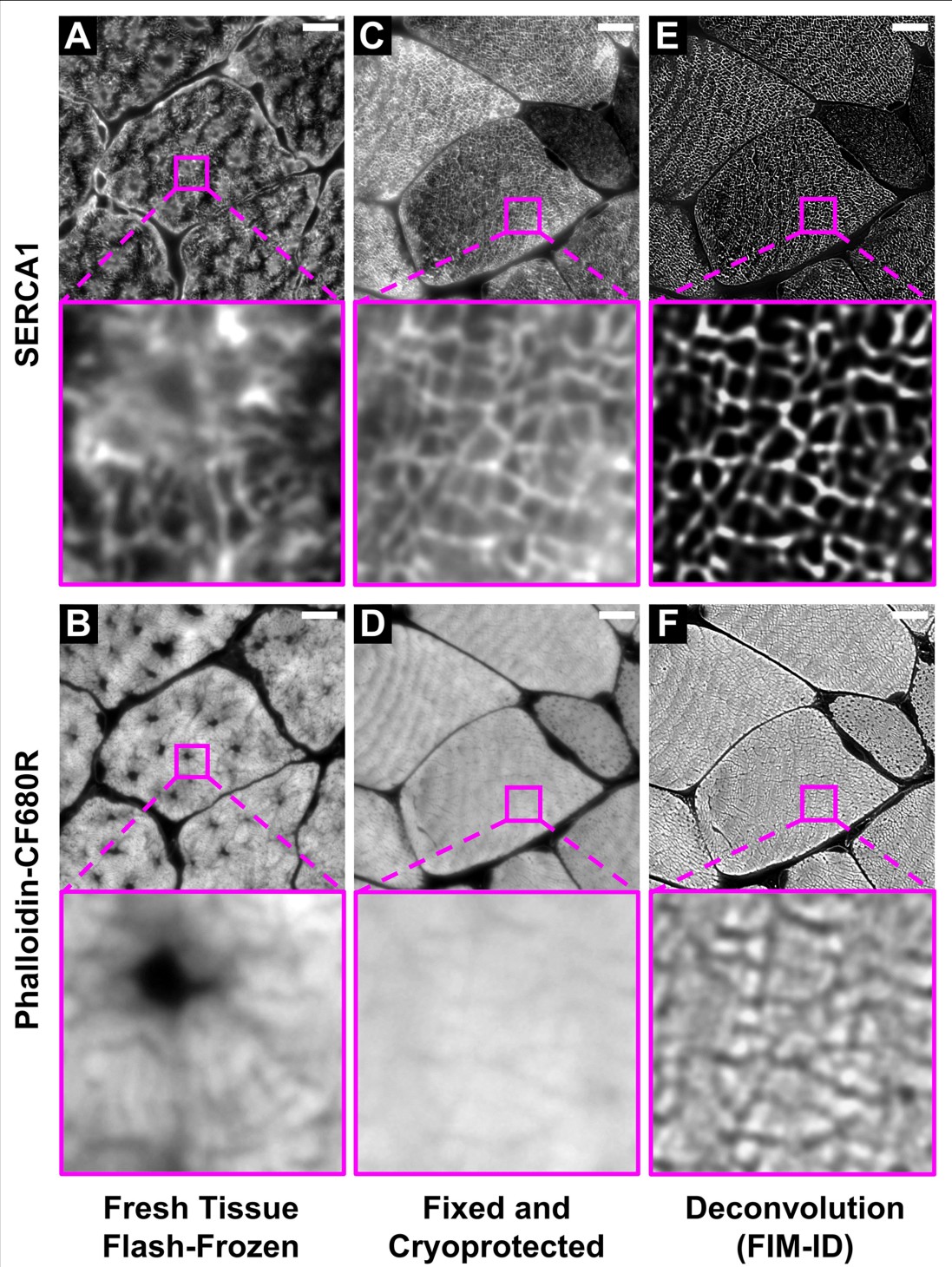

**Figure 1.** Optimization of the fixation, cryoprotection, and sectioning conditions for fluorescence imaging of myofibrils with image deconvolution (FIM-ID). Plantaris muscles from mice were collected and either (**A–B**) immediately flash-frozen and sectioned at –20°C, or (**C–D**) subjected to optimized fixation, cryoprotection, and sectioning conditions. Cross-sections of the muscles were stained with phalloidin-CF680R to identify F-actin and SERCA1 was immunolabeled with Alexa 594 to identify the periphery of the myofibrils. Images of SERCA1 (top) and F-actin (bottom) were captured from identical regions with a widefield fluorescence microscope. (**E–F**) The images in C–D were subjected to deconvolution. Scale bars = 10 µm.

The online version of this article includes the following figure supplement(s) for figure 1:

**Figure supplement 1.** Overview of the fluorescence imaging of myofibrils with image deconvolution (FIM-ID) workflow.

a time-effective method. However, it is known that the resolution and contrast in widefield images can be negatively impacted by the acquisition of light that resides outside of the focal plane. Fortunately, major advancements in deconvolution algorithms have made it possible to reassign the out-of-focus light to its appropriate location (*Stemmer et al., 2008*). Moreover, it has been shown that deconvoluted widefield images can actually have better resolution and contrast than confocal images (*Shaw, 2006*). Thus, to improve the resolution and contrast of the widefield images, we employed Leica's 'Computational Clearing' algorithm for image deconvolution (*Schumacher and Bertrand, 2019*). As shown in *Figure 1E–F*, the application of the deconvolution algorithm led to a dramatic increase in the resolution and contrast. Thus, by applying our optimized fixation, cryoprotection, and sectioning conditions, along with widefield fluorescence microscopy and image deconvolution, we fulfilled our goal of developing a time- and cost-effective method for visualizing intact myofibrils with a high degree of resolution and contrast, and we refer to the collective method as fluorescence imaging of myofibrils with image deconvolution (FIM-ID) (*Figure 1—figure supplement 1*).

## Validation of an automated pipeline for measuring myofibril size with FIM-ID

Having established the conditions for FIM-ID, we next sought to determine whether the images from FIM-ID would be amenable to automated measurements of myofibril size. To accomplish this, we first developed an automated pipeline in CellProfiler that could distinguish the periphery of the myofibrils (i.e. the SERCA signal) from the background within a single muscle fiber, and then all objects that were >90% enclosed by the signal from SERCA were identified (*Figure 3A–C*). Approximately one-half of these objects appeared to be unseparated clusters of myofibrils and/or obliquely sectioned single myofibrils. Thus, additional filtering steps were added so that subsequent measurements of size were only performed on cross-sections of single circular/oval myofibrils that had a maximal aspect ratio of approximately 2.5:1 (*Figure 3D*).

To test the accuracy of the automated measurements, eight randomly selected regions of interest (ROIs) from a single fiber were subjected to measurements of myofibril CSA. The same ROIs were given to six independent investigators, and these investigators were asked to identify and manually trace the periphery of all single circular/oval myofibrils that had an aspect ratio of <2.5:1 and were >90% enclosed by the SERCA signal (*Figure 3E*). The manual measurements of myofibril CSA were derived from the investigator's traces and compared with the automated measurements. As illustrated in *Figure 3F*, the comparisons revealed that there was a very high degree of agreement between the automated and manually obtained measurements of myofibril CSA. Indeed, a highly significant correlation (R=0.9589, p=0.0002) was observed when the automated measurements of myofibril CSA from each ROI were compared with the mean of the manual measurements. Thus, the results from these studies established that our automated program could accurately measure the size of myofibrils as identified with FIM-ID.

## Further refinement and validation of the automated measurements with FIM-ID

EM has been extensively used to assess the ultrastructural characteristics of skeletal muscle and numerous EM-based studies have reported measurements of myofibril size (*Wang et al., 1993*; *Ashmore and Summers, 1981*; *Goldspink, 1970*; *Penman, 1969*). Hence, to further test the validity of our method, we directly compared our automated FIM-ID measurements of myofibril CSA with manual EM-based measurements. Specifically, we performed an experiment with four mice in which one plantaris muscle from each mouse was subjected to FIM-ID, and the other was processed for EM-based imaging. With FIM-ID it was determined that all fibers in the plantaris muscles were SERCA1 positive, but during the EM imaging it became apparent that these SERCA1 positive fibers consisted of two clearly discernable fiber types. One type was generally large in CSA and possessed very few intermyofibrillar mitochondria, and we referred to these fibers as glycolytic (Gly) (*Figure 4A*). The other type was generally small in CSA and highly enriched with intermyofibrillar mitochondria, and we referred to these fibers as oxidative (Ox) (*Figure 4A*). The classification of the two fiber types was important because previous studies have shown that the morphological arrangement of myofibrils in different fiber types can be quite distinct. For instance, previous studies have shown that myofibrils in fibers that are rich in mitochondria typically have a 'felderstruktur' appearance (i.e. field structure) whereas

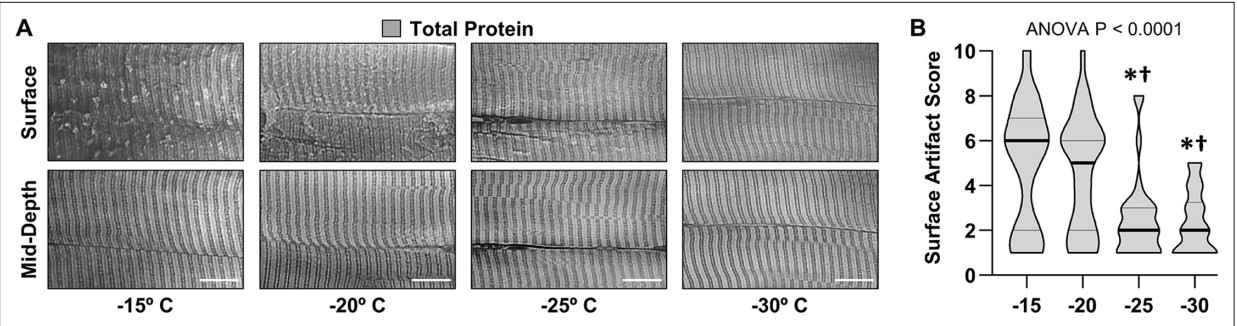

**Figure 2.** The effect of sectioning temperature on surface artifacts. An EDL muscle from a mouse was fixed with 4% paraformaldehyde, cryoprotected with 45% sucrose, and longitudinally sectioned at −15, −20, −25, or −30°C. The sections were stained for total protein and then five randomly selected regions of interest (ROIs) were imaged. Each fiber within the ROIs was scored for the presence of artifacts on the surface (a score of 1 indicates no artifacts and a score of 10 indicates extensive distortions). (**A**) Representative images from the surface and mid-depth of the sections. (**B**) Violin plot of the surface scores for the fibers at each of the different sectioning temperatures. Thick bars represent the median and thin bars represent the quartiles, n=28–39 fibers/group. Data was analyzed with one-way ANOVA. Significantly different from, * −15°C, † −20°C. Scale bars = 10 μm. Data used to generate B is provided in *Figure 2—source data 1*.

The online version of this article includes the following source data for figure 2:

**Source data 1.** Source data used to generate *Figure 2B*.

myofibrils in fibers that lack extensive mitochondria present with a 'fibrillenstruktur' appearance (i.e. fibril structure) (*Kruger and Gunther, 1955*; *Shear and Goldspink, 1971*; *Cheng and Breinin, 1965*; *Durston, 1974*). Accordingly, our experimental design was modified so that FIM-ID and EM-based measurements of myofibril CSA could be directly compared in the Ox and Gly fiber types.

To begin our comparisons, the average CSA of the myofibrils in the EM images of randomly selected Ox and Gly fibers was manually assessed as detailed in the Materials and methods. For the FIM-ID workflow, an additional step was needed to classify randomly selected fibers as being Ox or Gly, and this was done by taking advantage of previous studies which have shown that mitochondria are enriched with NADH and FAD+, which are endogenous fluorophores that can be excited with blue light (*Gooz and Maldonado, 2023*; *Schaefer et al., 2019*; *Kolenc and Quinn, 2019*; *Shadiow et al., 2023*). Specifically, as shown in *Figure 4B*, excitation of the FIM-ID samples with blue light yielded a weak autofluorescent decoration of the periphery of the myofibrils in all fibers, along with a prominent punctate signal in a subset of the fibers. The strongly autofluorescent fibers were generally small in CSA and closer examination of the autofluorescent signal revealed that the puncta were positioned at points in which there was a gap in the SERCA signal (*Figure 4—figure supplement 1*). It has been reported that intermyofibrillar mitochondria do not contain SERCA (*Jorgensen et al., 1982*), and therefore, the presence of the autofluorescent signal at sites in which there were gaps in the signal for SERCA, the small CSA of the fibers, and the pre-existing knowledge about the fluorescence properties of NADH and FAD+, all suggested that the autofluorescence was coming from the mitochondria. Accordingly, the strong autofluorescent fibers were classified as being equivalent to the Ox fibers identified with EM, and the non-autofluorescent fibers were classified as Gly fibers.

Our work with the autofluorescence signal not only provided us with a means for classifying the Ox vs. Gly fibers, but it also helped us realize that any myofibrils that are heavily surrounded by intermyofibrillar mitochondria would not meet the morphological criteria for automated measurements of CSA (i.e. the myofibrils would not present as objects that were >90% enclosed by the signal for SERCA). Thus, to address this, the automated measurements were performed on images in which the signal for autofluorescence and SERCA had been merged. With this approach, a more comprehensive decoration of the periphery of the myofibrils was obtained, and therefore, all of the remaining automated measurements on mouse skeletal muscle were conducted on merged images.

Having established the finer details of our workflow, we were finally able to compare automated measurements of myofibril CSA obtained with FIM-ID with the manual EM-based measurements. As shown in *Figure 4C*, the FIM-ID- and EM-based measurements both revealed the linear correlation between myofibril CSA and fiber CSA that has been reported in previous studies (*Jorgenson et al., 2020*; *Goldspink, 1970*), and there was no significant difference in the slope of this correlation when

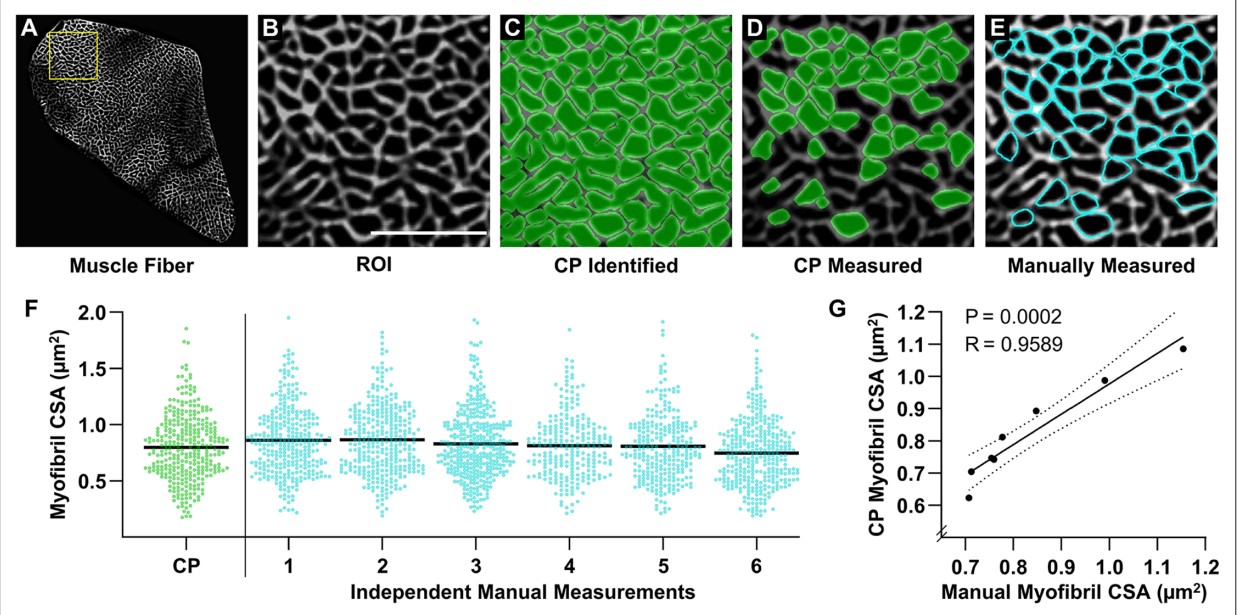

**Figure 3.** Validation of an automated pipeline for measuring myofibril size with fluorescence imaging of myofibrils with image deconvolution (FIM-ID). A cross-section from a mouse plantaris muscle was subjected to FIM-ID, and then eight regions of interest (ROIs) from a fiber (**A**) were subjected to automated and manual measurements of myofibril size. (**B**) Representation of an ROI from the fiber in A, scale bar = 5 µm. (**C**) Example of how the automated CellProfiler (CP) pipeline identifies the myofibrils (green). (**D**) Illustration of the myofibrils in CP that met the morphological criteria for subsequent measurements of cross-sectional area (CSA). (**E**) Example of the same ROI in B–D that was manually assessed for myofibrils that met the morphological criteria for subsequent measurements of CSA (cyan). (**F**) Scatter plot of the CSA of all myofibrils that were automatically measured by CP, or manually measured by independent investigators (n=6 investigators). The black bars represent the mean for each group. (**G**) For each ROI, the mean myofibril CSA as determined by CP was compared with the mean myofibril CSA from all of the manual measurements. The solid line represents the best fit from linear regression, the dashed lines represent the 95% confidence intervals, R indicates Pearson's correlation coefficient, and p indicates the likelihood that the relationship is significantly different from zero. Data used to generate F–G is provided in *Figure 3—source data 1*.

The online version of this article includes the following source data for figure 3:

**Source data 1.** Source data used to generate *Figure 3F–G*.

the two methods were compared. Moreover, no significant differences in mean myofibril CSA to fiber CSA ratio were found when the results from the FIM-ID and EM-based measurements were directly compared, and this point was true when the data from all analyzed fibers were considered, as well as when the results were separated according to the Ox and Gly fiber type (*Figure 4E*). Thus, it can be concluded that the automated measurements of myofibril CSA obtained with FIM-ID were indistinguishable from the manually obtained EM-based measurements.

As described in the Introduction, another goal of our study was to develop a time- and cost-effective method for measuring the number of myofibrils per fiber. With the EM images, this number was estimated for each fiber by manually obtaining measurements of the average myofibril CSA, the fiber CSA, and the percentage of the fiber CSA that was occupied by myofibrils. This was a time-intensive process, but with the latter two values, the total CSA of the fiber that was occupied by myofibrils could be calculated. The number of myofibrils per fiber was then derived by dividing the total CSA of the fiber that was occupied by myofibrils by the average myofibril CSA for that fiber. The same principles were also used to calculate the number of myofibrils per fiber with FIM-ID, but in this case, the total CSA of the fiber that was occupied by myofibrils was measured with an automated pipeline in CellProfiler. As shown in *Figure 4D*, the FIM-ID and EM-based measurements both revealed the linear correlation between myofibril number and fiber CSA that has been reported in previous studies (*Jorgenson et al., 2020*; *Goldspink, 1970*), and there was no significant difference in the slope of this correlation when the two methods were compared. Moreover, no significant differences in the mean number of myofibrils per fiber CSA were found when the results from the FIM-ID and EM-based measurements were directly compared, and this point was true when the data from all analyzed fibers were considered, as well as when the results were separated according to the Ox and Gly fiber type

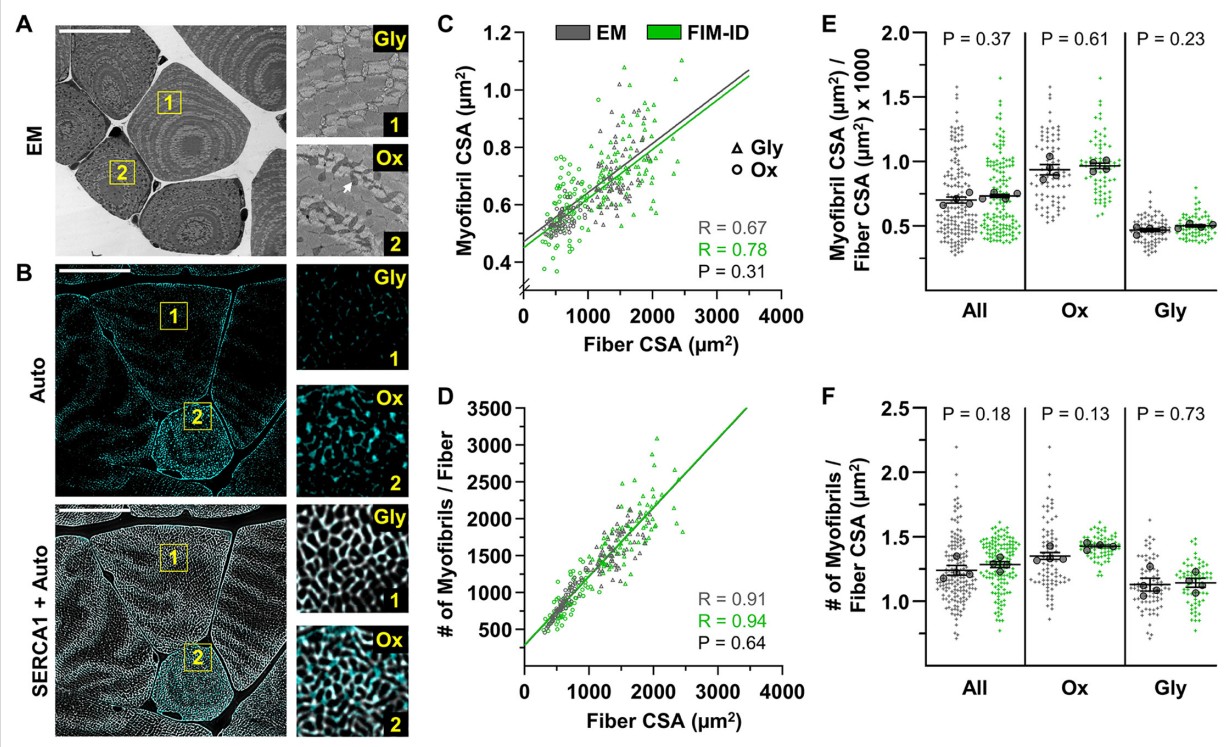

**Figure 4.** Automated measurements of myofibril size and number with fluorescence imaging of myofibrils with image deconvolution (FIM-ID) versus manual measurements with electron microscopy (EM). For each mouse, one plantaris muscle was (**A**) processed for imaging with EM, and the contralateral plantaris was (**B**) processed for FIM-ID, n=4 mice, scale bars = 25 μm. In the EM images, glycolytic (Gly) fibers and oxidative (Ox) fibers were distinguished by the presence of extensive intermyofibrillar mitochondria (arrow). In FIM-ID, the autofluorescence signal (Auto) was used to distinguish the Gly vs. Ox fibers. The 'Auto' signal was also merged with the signal from SERCA1 to provide a comprehensive decoration of the periphery of the myofibrils. (**C–D**) Scatter plots that illustrate the relationships between (**C**) myofibril cross-sectional area (CSA) vs. fiber CSA, and (**D**) the number of myofibrils per fiber vs. fiber CSA. The automated measurements from the FIM-ID samples are shown in green and the manual measurements from the EM samples are shown in gray. Individual values are presented as triangles for the Gly fibers and circles for the Ox fibers, n=144–160 fibers/condition (36–40 fibers/muscle). The solid lines represent the best fit from linear regression for each condition, R indicates Pearson's correlation coefficient, and p indicates the likelihood of a significant difference between the automated FIM-ID and the manual EM measurements. (**E–F**) The data in C–D was used to compare ratios of (**E**) the myofibril CSA to fiber CSA, and (**F**) the myofibril number to fiber CSA. The data is provided for all of the analyzed fibers and also separated according to fiber type. The large dots indicate the mean for each muscle. The data in C and D were analyzed with extra-sums-of-squares F-tests and the data in E and F were analyzed with paired t-tests or a Wilcoxon-matched paired t-test when the test for normality failed (i.e. the EM measurements of Ox fibers in F). Data used to generate C–F is provided in *Figure 4—source data 1*.

The online version of this article includes the following source data and figure supplement(s) for figure 4:

**Source data 1.** Source data used to generate *Figure 4C–F*.

**Figure supplement 1.** Punctate sites of autofluorescence align with gaps in the signal for SERCA1.

**Figure supplement 2.** Manual tracing of myofibrils from images acquired using electron microscopy.

**Figure supplement 3.** Using CellProfiler to determine the area occupied by the intermyofibrillar components.

---

(*Figure 4F*). Hence, just like myofibril CSA, the automated measurements of the number of myofibrils per fiber with FIM-ID were indistinguishable from the manually obtained EM-based measurements.

## The radial growth of muscle fibers that occurs in response to chronic mechanical overload is largely mediated by myofibrillogenesis

Having validated the accuracy of the FIM-ID measurements, we then set out to determine whether the radial growth of muscle fibers (i.e. an increase in fiber CSA) that occurs in response to an increase in mechanical loading is mediated by myofibril hypertrophy and/or myofibrillogenesis. We first addressed this in mice by subjecting their plantaris muscles to a chronic mechanical overload (MOV) or a sham surgical procedure. After a 16-day recovery period, the plantaris muscles were collected and processed for FIM-ID (*Figure 5B*). As shown in *Figure 5C*, the outcomes revealed that MOV led

to a 28% increase in fiber CSA when all fibers in the muscle were considered as a single group, and fiber type-specific analyses revealed that this was mediated by a 39% increase in the CSA of Ox fibers, and a 25% increase in the CSA of the Gly fibers. Similarly, the area per fiber that was occupied by myofibrils increased by 29% for all fibers, 39% for the Ox, and 26% for the Gly (*Figure 5D*). Moreover, the area per fiber that was occupied by the intermyofibrillar components (e.g. SR, mitochondria, etc.) also showed a 26% increase for all fibers, a 38% increase in the Ox fibers, and a 22% increase in the Gly fibers. When taken together, these results indicate that the MOV-induced increase in fiber CSA was mediated by a proportionate increase in the area that is occupied by the myofibrils and the area that is occupied by the intermyofibrillar components.

Next, we determined whether the increase in the area per fiber that was occupied by myofibrils was mediated by an increase in the size and/or number of myofibrils. Specifically, we first examined whether MOV altered the CSA of the myofibrils and, as shown in *Figure 5F*, the outcomes revealed that MOV increased the CSA of the individual myofibrils by 9% when all fibers were considered as a single group, and fiber type-specific analyses revealed that MOV led to a 14% increase in the Ox fibers, but no significant difference was detected in the Gly fibers (p=0.35). We then assessed the effect that MOV had on the number of myofibrils per fiber, and as shown in *Figure 5G*, it was concluded that MOV led to a 20% increase in the number of myofibrils per fiber when all fibers were considered as a single group, and fiber type-specific analyses revealed a 22% increase in the Ox fibers and a 21% increase in the Gly fibers. Importantly, during CellProfiler processing, objects in the sham and MOV samples were filtered from analysis at a very similar rate (44% vs 46%, respectively), and as such, the differences between the groups could not be explained by overt differences in the morphological properties of the myofibrils. Thus, the collective results of these analyses indicate that the MOV-induced increase in fiber CSA could be attributed to both myofibril hypertrophy and myofibrillogenesis, with the most robust and consistent effect occurring at the level of myofibrillogenesis.

## The radial growth of muscle fibers that occurs in response to progressive RE is largely mediated by myofibrillogenesis

To further address whether the radial growth of muscle fibers that occurs in response to an increase in mechanical loading is mediated by myofibril hypertrophy and/or myofibrillogenesis, we examined vastus lateralis muscle biopsies from humans that were collected before (Pre) and after (Post) they had performed 7 weeks of progressive RE (*Figure 6A*). As detailed in the Materials and methods, the samples were processed for FIM-ID with a step that allowed for the identification of SERCA1 vs. SERCA2 positive fibers (i.e. Type II vs. Type I fibers, respectively) (*Figure 6B*). As shown in *Figure 6C*, the outcomes revealed that RE led to an 18% increase in the CSA of SERCA1 positive fibers but did not significantly alter the CSA of SERCA2 positive fibers. Similar results were also observed when the SERCA1 and SERCA2 positive fibers were examined for changes in the area per fiber that was occupied by the myofibrils, as well as the area per fiber that was occupied by the intermyofibrillar components (*Figure 6D–E*). Specifically, SERCA1 positive fibers revealed an 18% and 17% increase in these values, respectively, while no significant differences were observed in SERCA2 positive fibers.

Thus, just like MOV, the RE-induced increase in the CSA of the SERCA1 positive fibers was mediated by a proportionate increase in the area that is occupied by the myofibrils and the area that is occupied by intermyofibrillar components.

To determine whether the increase in the area per fiber that was occupied by myofibrils was mediated by myofibril hypertrophy and/or myofibrillogenesis, we first examined whether RE altered the CSA of the myofibrils. As shown in *Figure 6F*, we detected a trend for a main effect of RE on myofibril CSA, but direct Pre vs. Post comparisons in the SERCA1 and SERCA2 positive fibers were not significant (p=0.31 and p=0.38, respectively). On the other hand, RE induced a clear fiber type-specific increase in the number of myofibrils per fiber with SERCA1 positive fibers revealing a 12% increase (p=0.009) while no significant difference was observed in SERCA2 positive fibers. It should also be noted that, during CellProfiler processing, objects in the Pre and Post samples were filtered from analysis at a very similar rate (47% vs 47%, respectively), and thus, the differences between the Pre and Post samples could not be explained by overt differences in the morphological properties of the myofibrils. As such, it can be concluded that the effects of RE were very similar to the effects of MOV in that the RE-induced increase in fiber CSA was largely mediated by the induction of myofibrillogenesis.

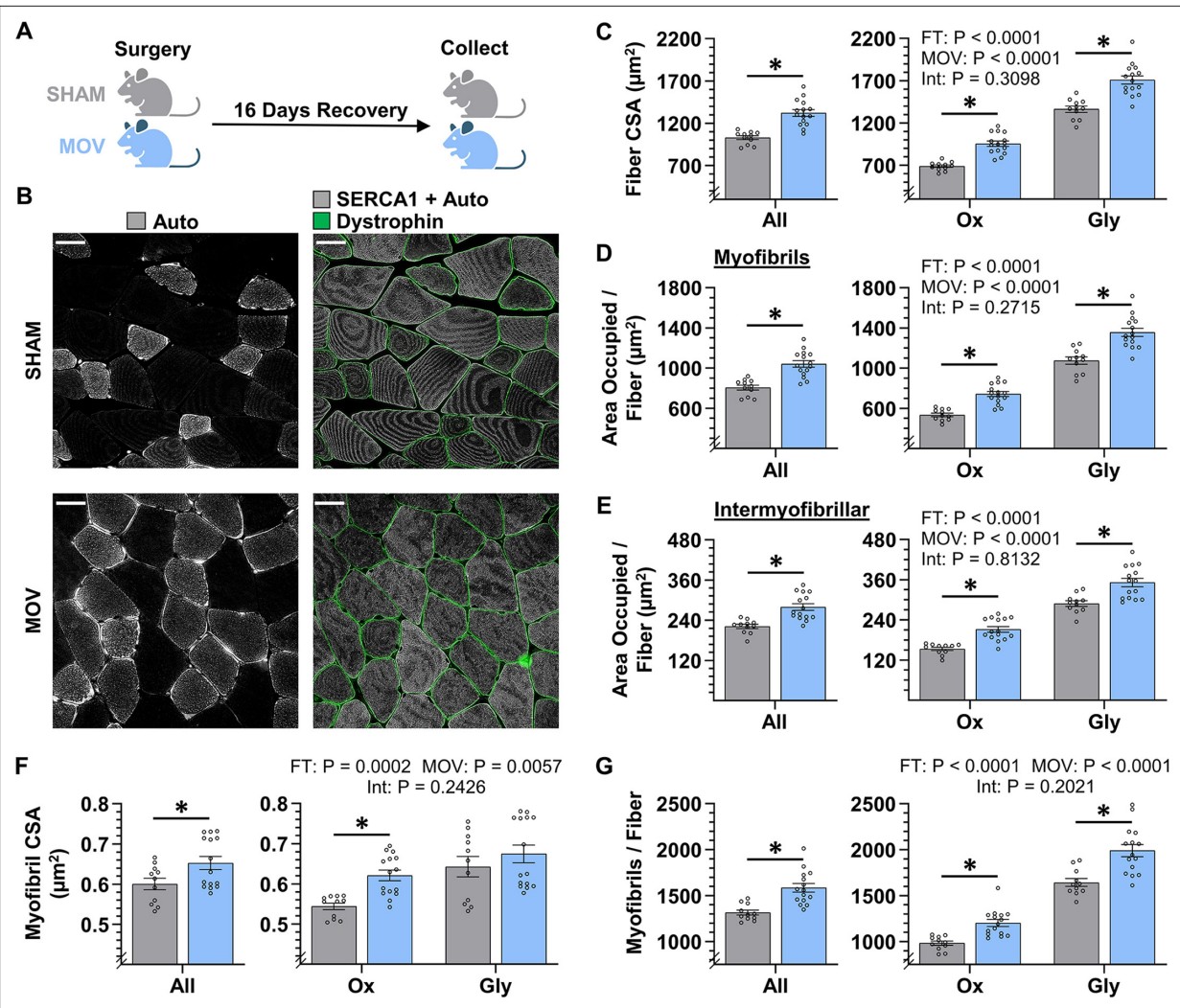

**Figure 5.** The radial growth of fibers that occurs in response to chronic mechanical overload (MOV) is largely mediated by myofibrillogenesis. (**A**) The plantaris muscles of mice were subjected to a chronic MOV or sham surgery, allowed to recover for 16 days, and then mid-belly cross-sections were subjected to fluorescence imaging of myofibrils with image deconvolution (FIM-ID). (**B**) Immunolabeling for dystrophin was used to identify the periphery of muscle fibers, autofluorescence (Auto) was used to distinguish glycolytic (Gly) fibers from highly oxidative (Ox) fibers, and the signal from SERCA1 plus Auto was merged and used to identify the periphery of the myofibrils, scale bars = 25 µm. (**C–G**) Graphs contain the results from all of the fibers that were analyzed, as well as the same data after it was separated according to fiber type. (**C**) Fiber cross-sectional area (CSA), (**D**) the area per fiber occupied by myofibrils, (**E**) the area per fiber occupied by intermyofibrillar components, (**F**) myofibril CSA, and (**G**) the number of myofibrils per fiber. The data are presented as the mean ± SEM, n=11–15 muscles/group (All = 24–54 fibers/muscle, Ox and Gly = 10–24 fibers/muscle, Gly = 10–24 fibers/muscle, and an average of 356±17 myofibrils/fiber). Student's t-tests were used to analyze the data in the 'All' graphs and two-way ANOVA was used to analyze the data in the fiber-type graphs. Insets show the p values for the main effects of MOV, fiber type (FT), and the interaction (Int). * Significant effect of MOV, p<0.05. Graphic illustration in A was created with BioRender.com. Data used to generate C–G is provided in *Figure 5— source data 1*.

The online version of this article includes the following source data and figure supplement(s) for figure 5:

**Source data 1.** Source data used to generate *Figure 5C–G* and *Figure 5—figure supplement 1*.

**Figure supplement 1.** The radial growth of fibers that occurs in response to an increase in mechanical loading is correlated with the induction of myofibrillogenesis.

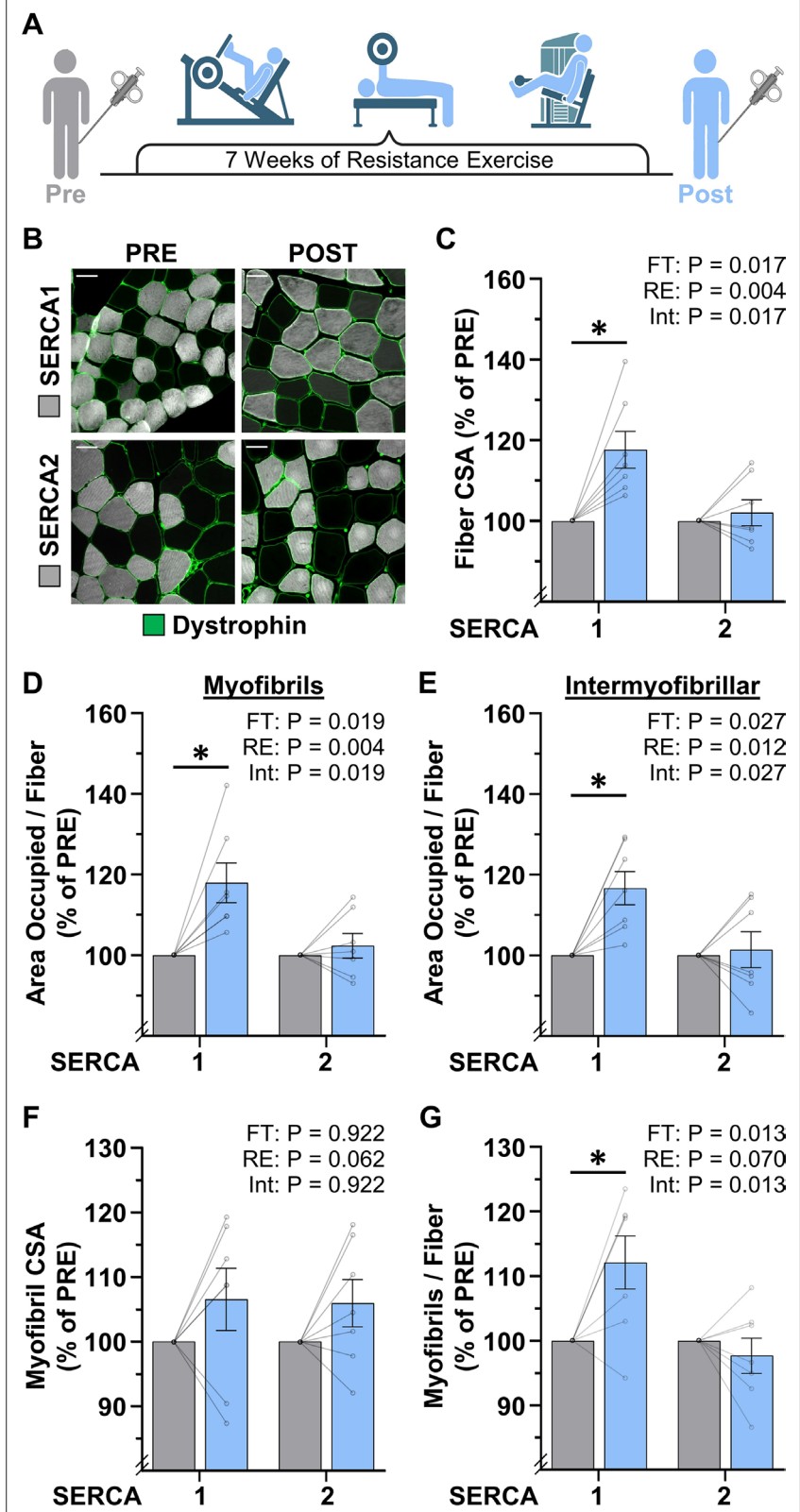

**Figure 6.** The radial growth of fibers that occurs in response to resistance exercise (RE) is largely mediated by myofibrillogenesis. (**A**) Biopsies of the vastus lateralis were collected before (PRE) and after (POST) participants performed 7 weeks of progressive RE. (**B**) Cross-sections were immunolabeled for dystrophin (to identify the periphery of muscle fibers), SERCA1 (to identify the periphery of the myofibrils in Type II fibers), or SERCA2 (to

*Figure 6 continued on next page*

*Figure 6 continued*

identify the periphery of the myofibrils in Type I fibers), and subjected to fluorescence imaging of myofibrils with image deconvolution (FIM-ID), scale bars = 50 μm. (**C–G**) Graphs contain the values for each subject expressed relative to their PRE sample. (**C**) Fiber cross-sectional area (CSA), (**D**) the area per fiber occupied by myofibrils, (**E**) the area per fiber occupied by intermyofibrillar components, (**F**) myofibril CSA, and (**G**) the number of myofibrils per fiber. The data are presented as the mean ± SEM, n=7 participants (SERCA1 15–33 fibers/participant, SERCA2 8–38 fibers/participant, and an average of 1101±60 myofibrils/fiber). Significance was determined by repeated measures two-way ANOVA. Insets show the p values for the main effects of RE, fiber type (FT), and the interaction (Int). * Significant effect of RE, p<0.05. Graphic illustration in A was created with BioRender.com. Data used to generate C–G is provided in *Figure 6—source data 1*.

The online version of this article includes the following source data and figure supplement(s) for figure 6:

**Source data 1.** Source data used to generate *Figure 6C–G* and *Figure 6—figure supplement 1*.

**Figure supplement 1.** The radial growth of fibers that occurs in response to an increase in mechanical loading is correlated with the induction of myofibrillogenesis.

## Discussion

In this study, we developed FIM-ID as a means for visualizing myofibrils with a fluorescence microscope. As illustrated throughout the study, the images from FIM-ID have a high degree of resolution and contrast, and these properties enabled us to develop pipelines that can perform automated measurements of myofibril size and myofibril number. Importantly, these pipelines were developed in a free open-source program (CellProfiler), and our manuscript includes copies of the pipelines along with instructions and test images so that prospective users can easily employ them (see *Source code 1*). Moreover, the bulk of the FIM-ID workflow utilizes common lab instruments, and essentially anyone who has access to a standard cryostat and a fluorescent microscope with a motorized Z-stage can perform it. Indeed, for first-time users, we suspect that the most complicated part of the workflow will be the implementation of image deconvolution. Fortunately, deconvolution modules have become a commonly available add-on for commercial imaging platforms, and when not available, there are several open-source deconvolution algorithms that users could deploy (*Lam et al., 2017*; *Makarkin and Bratashov, 2021*; *Katoh, 2019*; *Su et al., 2023*). As such, we are confident that a broad range of investigators will be able to take advantage of our method.

With the development of our method, we were able to gain insight into whether the radial growth of muscle fibers that occurs in response to an increase in mechanical loading is mediated by myofibril hypertrophy and/or myofibrillogenesis. This is a fundamentally important question in the field of skeletal muscle growth and therefore we addressed it with two model systems. As expected, both models induced significant radial growth of the SERCA1 positive muscle fibers. However, a significant increase in myofibril CSA (i.e. myofibril hypertrophy) was not consistently observed (*Figures 5–6*), and no significant correlation between the magnitude of the radial growth of the fibers and myofibril hypertrophy was detected in any of the fiber types that we examined (*Figure 5—figure supplement 1*, *Figure 6—figure supplement 1*). At face value, these results could be viewed as evidence that myofibril hypertrophy does not play a major role in the radial growth response. However, it is important to consider that the results were obtained from muscles in which the radial growth response was already well advanced. This is important because, according to Goldspink's model of myofibril splitting (*Goldspink, 1970*; *Goldspink, 1971*; *Patterson and Goldspink, 1976*; *Goldspink, 1983*), myofibril hypertrophy would only be detected during the early stages of the radial growth response (i.e. before the induction of myofibril splitting which would lead to a reduction in the CSA of the involved myofibrils). Thus, to further address the role of myofibril hypertrophy, future studies should obtain data from muscles that are going through various stages of the radial growth response (i.e. a time course). With the use of FIM-ID, the pursuit of such studies would be a feasible endeavor and the resulting data would not only help to resolve whether myofibril hypertrophy contributes to the radial growth response, but it would also provide insight into whether myofibril hypertrophy precedes the onset of myofibrillogenesis as predicted by the myofibril splitting model.

In contrast to the induction of myofibril hypertrophy, all the fiber types in this study that showed significant radial growth also showed a significant increase in the number of myofibrils per fiber (i.e.

myofibrillogenesis) (*Figures 5–6*). Moreover, each of these fiber types revealed a significant positive correlation between the magnitude of radial growth and the magnitude of myofibrillogenesis (*Figure 5—figure supplement 1*, *Figure 6—figure supplement 1*). These points, when coupled with the other data presented in *Figures 5 and 6*, indicate that radial growth of the muscle fibers was largely mediated by myofibrillogenesis. Importantly, this conclusion was upheld in two very distinct models of increased mechanical loading (i.e. chronic MOV in mice and progressive RE in humans). Thus, it appears that the induction of myofibrillogenesis is a conserved component of the radial growth response.

To the best of our knowledge, this is the first study to demonstrate that an increase in mechanical loading can induce myofibrillogenesis. Moreover, the results of this study indicate that myofibrillogenesis is the major driver of the radial growth response. Given these conclusions, one is faced with the question of how an increase in mechanical loading induces myofibrillogenesis. As mentioned several times, one possibility is that the induction of myofibrillogenesis is mediated by hypertrophy and subsequent splitting of the pre-existing myofibrils (i.e. the myofibril splitting model) (*Goldspink, 1970*; *Goldspink, 1971*; *Patterson and Goldspink, 1976*; *Goldspink, 1983*). However, numerous studies have also described a different type of myofibrillogenesis that involves the de novo formation of myofibrils (*Wang et al., 2022*; *Fenix et al., 2018*; *Rhee et al., 1994*; *Sanger et al., 2002*). Specifically, with de novo myofibrillogenesis, it is thought that new myofibrils are derived from nascent proteins and that these nascent proteins are assembled into new myofibrils via a three-step sequence of events (for more details, please refer to *Wang et al., 2022*; *Fenix et al., 2018*). Importantly, the concepts of de novo myofibrillogenesis have largely been studied within the context of myogenesis (i.e. during the early stages of differentiation, embryonic development, etc.), but whether de novo myofibrillogenesis can be induced in adult skeletal muscle remains largely unexplored. In our opinion, the concepts of myofibril splitting and de novo myofibrillogenesis are very intriguing, and we consider both models to be worthy of further investigation.

In summary, the outcomes of this study have revealed that the induction of myofibrillogenesis plays a major role in the mechanically induced growth of skeletal muscle. The mechanisms via which an increase in mechanical loading induces myofibrillogenesis are not known, but the groundwork for two different models has already been established. Studies aimed at testing the validity of these models should lead to important advancements in the field, and with the advent of FIM-ID, such studies can now be performed in a time- and cost-effective manner.

## Materials and methods
### Animals and ethical approval

Experimental procedures were performed on male C57BL/6J mice (Jackson Laboratories) that were 8–10 weeks of age. All animals were housed in a room maintained at 25°C with a 12 hr light/dark cycle and received rodent laboratory chow (Purina) and water ad libitum. Before all surgical procedures, mice were anesthetized with 1–5% isoflurane mixed in oxygen at a flow rate of 1.0 L/min, which was maintained throughout the surgery. Immediately following the completion of the surgery, the mice were given an intraperitoneal injection of 0.05 µg/g of buprenorphine in phosphate-buffered saline (PBS). During collection, the mice were euthanized by cervical dislocation under anesthesia before removing the hindlimbs. The Institutional Animal Care and Use Committee (IACUC) at the University of Wisconsin-Madison approved all the methods used in this study under protocol #V005375.

### Chronic MOV

Bilateral chronic MOV surgeries were performed with a modified version of our previously described procedure (*You et al., 2019*). Specifically, the plantaris muscles were subjected to MOV by removing the distal one-third of the gastrocnemius muscle while leaving the soleus and plantaris muscle intact. Mice in the control groups were subjected to a sham surgery where an incision was made on the lower leg and then closed. Following the surgeries, incisions were closed with nylon non-absorbable sutures (Oasis) and Vetbond Tissue Adhesive (3M). After 16 days of recovery, the plantaris muscles were collected and processed as described below.

## Collection and processing of mouse skeletal muscles

The hindlimbs were collected by retracting the skin surrounding the entire leg and then the limb was severed at the hip joint. The foot and proximal end of the femur were secured onto a circular piece of aluminum wire mesh (Saint-Gobain ADFORS insect screen) with 90° angles between the foot-tibia and tibia-femur joints using 4-0 non-absorbable silk sutures (Fine Science Tools). The hindlimbs were then submerged in a glass beaker filled with 20 mL of 4% paraformaldehyde in 0.1 M phosphate buffer (PB) solution (25 mM sodium phosphate monobasic, 75 mM sodium phosphate dibasic, pH 7.2) for 3 hr at room temperature on a tabletop rocker set to 50 RPM. Extensor digitorum longus and plantaris muscles were then extracted from the hindlimb and submerged in a 1.5 mL Eppendorf tube that was filled with 1.0 mL of 4% paraformaldehyde in 0.1 M PB. The tubes were then placed on a nutating rocker for 21 hr at 4°C. For the subsequent cryoprotection, each muscle was submerged in a 1.5 mL Eppendorf tube filled with 1.0 mL of 15% sucrose in 0.1 M PB and placed back on a nutating rocker for 6 hr at 4°C, followed by immersion in 45% sucrose in 0.1 M PB for 18 hr at 4°C. Each muscle was then rapidly immersed in OCT (Tissue-Tek), frozen in liquid nitrogen-chilled isopentane, and stored at –80°C (*Hibbert et al., 2023*).

## Collection and processing of human skeletal muscle biopsies

Human biopsy samples were collected in conjunction with a recently published study by the Roberts and Kavazis laboratories (*Mesquita et al., 2023*). Briefly, healthy (BMI: 25.2±5 kg/m$^2$) young (23±4 years) male participants who had not performed organized resistance training over the previous 3 years were selected. Participants performed RE twice a week for 7 weeks, which included leg press, bench press, leg extension, cable pull-down, and leg curls. Both the training volume and load were progressively increased throughout the 7-week training program. Muscle biopsies were collected at the mid-belly of the vastus lateralis muscle before and 72 hr after completion of the 7 weeks of resistance training. Biopsies were fixed in 4% paraformaldehyde in 0.1 M PB for 24 hr, incubated in a 15% sucrose in 0.1 M PB solution for 6 hr, incubated in a 30% sucrose in 0.1 M PB solution for 18 hr at 4°C, immersed in OCT, and then frozen in liquid nitrogen-chilled isopentane, and stored at –80°C. For more details about participant selection and the resistance training regimen, please refer to *Mesquita et al., 2023*.

## Immunohistochemistry

Unless otherwise noted, mid-belly (mouse) and biopsy (human) cross-sections (5 µm thick) from muscles frozen in OCT were taken with a cryostat chilled to –30°C and collected on Superfrost Plus microscope slides (Fisher Scientific). Immediately upon collection, the sections were transferred to diH$_2$O water for 15 min to hydrate the samples. While ensuring that the sections remained hydrated, the area on the slide surrounding the section was dried with a Kimwipe, and then a hydrophobic circle was drawn around the section using an Aqua Hold 2 pen (Scientific Device Laboratory). Slides were then placed into a humidified box for all subsequent washing and incubation steps, which were all performed at room temperature on a rotating rocker set to 50 RPM.

### Primary antibody labeling

Sections were washed with PBS for 5 min and then incubated for 30 min in blocking solution (0.5% Triton X-100, 0.5% bovine serum albumin dissolved in PBS). For mouse samples, sections were incubated overnight in blocking solution containing mouse IgG1 anti-SERCA1 (1:100, VE121G9, Santa Cruz #SC-58287) and rabbit anti-dystrophin (1:100, Thermo Fisher #PA1-21011). For human biopsy samples, sections were incubated overnight in blocking solution containing mouse IgG1 anti-SERCA1 or mouse IgG1 anti-SERCA2 (1:200, Santa Cruz #SC-53010) and rabbit anti-dystrophin. Following the overnight incubation, sections were washed three times for 10 min with PBS, followed by three more washes for 30 min with PBS.

### Secondary antibody labeling

All sections labeled with primary antibodies were incubated overnight in blocking solution containing Alexa Fluor 594-conjugated goat anti-mouse IgG, Fcγ Subclass 1 specific (1:2000, Jackson ImmunoResearch #3115-585-205) and Alexa Fluor 488-conjugated goat anti-rabbit IgG (1:5000, Invitrogen #A11008). Following the overnight incubation, sections were washed three times for 10 min with PBS, followed by three more times for 30 min with PBS.

### Labeling with phalloidin

In some instances, after the secondary antibody labeling step, sections were incubated in blocking solution containing Phallodin-CF680R (1:50, Biotium #00048) for 20 min. Following the incubation, the sections were washed three times for 10 min with PBS, followed by three more times for 30 min with PBS.

### Mounting

Washed sections were mounted with 30 µL of ProLong Gold Antifade Mountant (Thermo Fisher #P36930) and covered with a 22×22 mm² No. 1 glass slip (Globe Scientific #1404-10). All sections were allowed to cure in the mountant for at least 24 hr while being protected from light before imaging.

## Assessment of muscle fiber size

The cross-sections that had been labeled for dystrophin and SERCA1/2 were imaged with a Keyence automated BZX700 inverted epifluorescence microscope via a 10× objective lens. For each fluorescent channel (FITC and TxRED filter), a 3×3 (mouse samples) or 5×5 (human samples) field was captured and stitched together using the Keyence Analyzer Software. The mean CSA of all qualified muscle fibers within the stitched images was then measured with our previously published CellProfiler pipeline (*Zhu et al., 2021*).

## Fluorescence imaging of myofibrils and image deconvolution

Randomly selected ROIs from the cross-sections that had been labeled for dystrophin and SERCA1/2 were imaged with an HC PL APO 63×/1.40 oil immersion objective on a THUNDER Imager Tissue 3D microscope (Leica). Images for dystrophin were captured through a GFP filter (EX: 470/40, EM: 525/50), SERCA1/2 through a TXR filter (EX: 560/40, EM: 630/76), and autofluorescence was captured through a 405 filter (EX: 405/60, EM: 470/40). Z-stacks with a step size of 0.22 µm were captured through the entire thickness of the sample. Deconvolution was then applied to all images in the Z-stack for each channel using Leica's Small Volume Computational Clearing (SVCC) algorithm with the refractive index set to 1.47, strength set to 80–100%, and regularization set to 0.05. The deconvoluted images were then opened in ImageJ and the Z-plane with the most in-focus image for SERCA1/2 was chosen. When appropriate, the autofluorescence signal from the same plane was merged with the image for SERCA1. This was done in ImageJ with the Z-Project function set to 'Sum Slices' and the merged image was saved as a single grayscale TIFF image. The single or merged images were then converted to a 16-bit TIFF file format and the pixel density was adjusted to 6144×6144 by using the ImageJ resize function with the following settings: 'Constrain aspect ratio', 'Average when downsizing', and 'Bilinear' interpolation.

## Total protein labeling and scoring for sectioning artifacts

Longitudinal sections of the EDL muscle (3 µm thick) were collected on a cryostat that was chilled at –15°C to –30°C and collected on Superfrost Plus microscope slides. The sections were stained for total protein with a No-Stain kit (Invitrogen #A44449). Specifically, 25 µL of the 20× No-Stain Labeling Buffer stock solution was diluted into 475 µL diH$_2$O to create a working 1× solution. Next, 1 µL of the No-Stain Activator was added to the 500 µL 1× solution and vortexed. Then, 1 µL of the No-Stain Derivatizer was added to the 1× solution and vortexed. Labeling was then performed by adding 50 µL of the prepared No-Stain solution to the sections, which were then incubated for 30 min at room temperature with rocking at 50 RPM. Labeled sections were then washed three times for 5 min with diH$_2$O. After washing, the diH$_2$O was removed, and 4 µL of ProLong Diamond mounting medium (Life Technologies #P36965) was added to the sections before covering with a 12 mm No. 1 glass slip (Neuvitro #GG1215H). After curing for 48 hr, randomly selected ROIs were imaged with a THUNDER Imager with a Y3 filter (Ex: 545/26, EM: 605/70) and subjected to deconvolution as described for FIM-ID. The images were then examined by a blinded investigator and each fiber within a given ROI was scored for the presence of artifacts on the surface of the section (a score of 1 indicates no artifacts and a score of 10 indicates extensive distortions).

## Automated measurements of myofibril size and number with FIM-ID

### Assessment of myofibril size and number in muscles from mice

Within each image from FIM-ID, individual fibers were randomly selected by a blinded investigator and manually traced in ImageJ to record the fiber CSA. Then, the traced fiber was isolated for automated analysis by choosing the 'Clear Outside' option. The resulting image was then cropped so that it only contained the isolated fiber. The cropped image was then saved as a .tiff file. For the studies in *Figures 4 and 5*, preliminary analyses were performed on 24 fibers per sample with an approximately equal number of autofluorescent and non-autofluorescent fibers. Additional ROIs/fibers were then randomly selected for analysis until the average CSA of the selected fibers fell within ±10% of the mean fiber CSA that was observed for the entire muscle during the Keyence-based analyses. For automated measurements of myofibril size, each single fiber image was imported and processed with our custom pipeline 'Myofibril CSA Analysis' in CellProfiler version 4.2.1, which measures the mean myofibril CSA for each fiber (*Stirling et al., 2021*). To calculate the number of myofibrils per fiber, the total CSA of the fiber that was occupied by myofibrils had to be determined. To obtain this value, each single fiber image that contained the merged signal for autofluorescence and SERCA1 was imported and processed with our custom pipeline 'Intermyofibrillar Area' in CellProfiler (*Figure 4— figure supplement 3*). This pipeline performs automated measurements of the area occupied by the merged autofluorescence and SERCA1 signal for each fiber (i.e. the area occupied by the intermyofibrillar components such as the SR, mitochondria, etc.). This value was then subtracted from the fiber CSA to obtain the total area occupied by the myofibrils for each fiber. Finally, the number of myofibrils per fiber was determined by dividing the total area occupied by the myofibrils by the mean myofibril CSA. Note: the 'Myofibril CSA Analysis' and 'Intermyofibrillar Area' pipelines as well as user instructions and test images have all been included as supplemental material.

### Assessment of myofibril size and number in muscles from humans

The same general methods that were used in mice were applied to the analysis of human samples. Exceptions to this included that the analyses were performed on images of SERCA1 or SERCA2. Furthermore, preliminary analyses were performed on 16 fibers (8 SERCA1, 8 SERCA2 fibers), and additional ROIs/fibers were selected for analysis until the average CSA of the selected fibers fell within ±4% of the mean fiber CSA that was observed with for the entire muscle section during the Keyence-based analyses.

## Processing of mouse skeletal muscle for EM

The hindlimbs of mice were removed and secured to an aluminum wire mesh as described in the section entitled 'Collection and processing of the mouse skeletal muscles'. The hindlimbs were then submerged in a 100 mL glass beaker filled with 20 mL of Karnovsky fixative (*Karnovsky, 1965*) (2.5% EM-grade glutaraldehyde [Electron Microscopy Sciences #16019] plus 2.0% EM-grade paraformaldehyde [Electron Microscopy Sciences #15714S] in 0.1 M PB) for 3 hr at room temperature on a tabletop rocker set to 50 RPM. Each plantaris muscle was then extracted from the hindlimb and submerged in a 1.5 mL Eppendorf tube containing 1.0 mL of 2.5% EM-grade glutaraldehyde plus 2.0% EM-grade paraformaldehyde in 0.1 M PB solution and placed on a nutating rocker for 21 hr at 4°C. The fixed plantaris muscles were then transversely cut at the mid-belly and further transversely cut into 1 mm thick slices on either side of the midline.

Unless otherwise noted, the following steps were performed at room temperature in approximately 10 mL of solution in 20 mL scintillation vials on a rotator set at 10 RPM. Tissue slices were first washed three times for 15 min with 0.1 M PB solution, followed by overnight incubation in 1% osmium tetroxide (Electron Microscopy Sciences #19170) in 0.1 M PB solution at room temperature. The samples were then washed four times for 5 min in 0.1 M PB solution, followed by graded dehydration in ethanol with 5 min incubation steps (2×35%, 2×50%, 2×70%, 2×80%, 2×90%, 2×95%, 4×100%). Samples were then incubated in 100% anhydrous acetone for 2×5 min and 2×10 min. Epon resin was prepared by mixing 6.25 g Epon 812, 3.25 g DDSA, 3.0 g NMA, and 8 drops of DMP 30 (Electron Microscopy Sciences #14120) before incubating the samples in a 3:1 acetone:epon solution overnight at room temperature. Samples were then incubated in a 1:1 acetone/epon solution for 2 hr, followed by a 1:3 acetone/epon solution for 2 hr, before being placed in 100% epon overnight in a

vacuum at room temperature. Samples were then added to fresh 100% epon for 3×30 min at 60°C before being placed into molds with fresh 100% epon to cure at 60°C overnight.

The resin-embedded samples were sectioned on a Leica EM UC6 ultramicrotome at 500 nm to generate semi-thin sections using a Histo Diamond Knife (DiATOME) and stained with Toluidine blue to confirm proper orientation (i.e. the cross-sections of the muscle fibers were oriented perpendicular to the length of the fibers). After confirming the correct orientation of the block, the samples were sectioned at 100 nm and collected on carbon formvar-coated copper grids (Electron Microscopy Sciences #S2010NOT). Sections were then contrast-stained with 8% uranyl acetate (Electron Microscopy Sciences #22400) in 50% ethanol for 10 min at room temperature, followed by lead citrate (Electron Microscopy Sciences #22410) for 10 min at room temperature with 4×30 s washes in diH$_2$O water after each contrast stain. The sections were viewed at 80 kV on a Philips CM120 transmission electron microscope equipped with AMT BioSprint12 digital camera.

## Manual measurements of myofibril size and number with EM

For each muscle processed for EM, 20 Ox and 20 non-Ox fibers were randomly selected for imaging. For each fiber, one low-magnification image (×1250) and one randomly chosen high-magnification region (×5600) within the fiber were captured. The low-magnification images were used to measure the fiber CSA and the high-magnification images were used to measure the myofibril CSA as well as the percentage of the fiber CSA that was occupied by myofibrils. Specifically, for each high-magnification image, ≈30 randomly selected myofibrils were manually traced and used to determine the mean myofibril CSA for that fiber. To estimate the percentage of the fiber CSA that was occupied by myofibrils, an ROI from the high-magnification image that had an equal representation of the A- and I-band regions of the myofibrils was selected. Then, within that ROI, all areas containing myofibrils were manually traced and the total area occupied by the myofibrils was recorded. The total CSA of the myofibrils in the ROI was then divided by the total area of the ROI to obtain the percentage of the fiber CSA that was occupied by the myofibrils. This value was then used to calculate the total CSA of fiber that was occupied by myofibrils. Finally, the number of myofibrils per fiber was determined by dividing the total CSA of the fiber that was occupied by myofibrils by its mean myofibril CSA. See *Figure 4—figure supplement 2* for an example of the process for manual measurements.

## Statistical analysis

Statistical significance was determined by using Student's t-tests, paired t-tests, Wilcoxon-matched paired t-tests, two-way ANOVA with Bonferroni post hoc comparisons, or repeated measures two-way ANOVA with Bonferroni post hoc comparisons, as indicated in the figure legends. Samples that deviated more than three times from the mean within a given group were excluded as outliers. Differences between groups were considered significant when $p < 0.05$. All statistical analyses were performed with GraphPad Prism version 10.0.0 for Windows (Dotmatics).

## Acknowledgements

The research reported in this publication was supported by the National Institute of Arthritis and Musculoskeletal and Skin Diseases of the National Institutes of Health (NIH) under Awards AR074932 and AR082816 to TAH, the National Institute of General Medical Sciences of the NIH under Award GM141739 to MCM, and a National Strength and Conditioning Association Foundation Doctoral Student Award to PHCM. The authors would like to thank Benjamin August and Randy Massey for their assistance with the EM, Nathaniel Steinert, Wenyuan Zhu, and Philip Flejsierowicz for their contributions to the validation of the CellProfiler pipeline, as well as Breanna Mueller, Casey Sexton, and Cleiton Libardi for their contributions to the human study.

# Additional information

## Funding

| Funder | Grant reference number | Author |
|---|---|---|
| National Institute of Arthritis and Musculoskeletal and Skin Diseases | AR074932 | Troy A Hornberger |
| National Institute of Arthritis and Musculoskeletal and Skin Diseases | AR082816 | Troy A Hornberger |
| National Institute of General Medical Sciences | GM141739 | Mason C McIntosh |
| National Strength and Conditioning Association | Doctoral Student Award | Paulo HC Mesquita |

The funders had no role in study design, data collection and interpretation, or the decision to submit the work for publication.

## Author contributions

Kent W Jorgenson, Conceptualization, Data curation, Software, Formal analysis, Validation, Investigation, Visualization, Methodology, Writing – original draft, Writing – review and editing; Jamie E Hibbert, Data curation, Formal analysis, Investigation, Writing – original draft, Writing – review and editing; Ramy KA Sayed, Anthony N Lange, Formal analysis, Investigation; Joshua S Godwin, Paulo HC Mesquita, Bradley A Ruple, Mason C McIntosh, Andreas N Kavazis, Investigation; Michael D Roberts, Supervision, Investigation, Writing – original draft, Project administration, Writing – review and editing; Troy A Hornberger, Conceptualization, Data curation, Formal analysis, Supervision, Funding acquisition, Validation, Investigation, Visualization, Methodology, Writing – original draft, Project administration, Writing – review and editing

## Author ORCIDs

Kent W Jorgenson ⓘ http://orcid.org/0000-0001-5206-8507
Jamie E Hibbert ⓘ https://orcid.org/0000-0001-7327-6460
Ramy KA Sayed ⓘ https://orcid.org/0000-0003-4467-6742
Troy A Hornberger ⓘ https://orcid.org/0000-0002-2349-1899

## Ethics

The current study was reviewed and approved by the Institutional Review Board at Auburn University (Protocol # 21-390 FB) and conformed to the standards of the Declaration of Helsinki, except that it was not registered as a clinical trial. All participants were informed of the procedures and risks of the current study before providing written consent.

The Institutional Animal Care and Use Committee (IACUC) at the University of Wisconsin-Madison approved all the methods used in this study under protocol #V005375.

Reviewer #1 (Public Review): https://doi.org/10.7554/eLife.92674.3.sa1
Reviewer #2 (Public Review): https://doi.org/10.7554/eLife.92674.3.sa2
Reviewer #3 (Public Review): https://doi.org/10.7554/eLife.92674.3.sa3
Author Response https://doi.org/10.7554/eLife.92674.3.sa4

# Additional files

## Supplementary files
• Source code 1. CellProfiler pipelines, instructions, and sample images.

• MDAR checklist

## Data availability

The "Myofibril CSA Analysis" and "Intermyofibrillar Area" CellProfiler pipelines as well as user instructions and test images have all been included as source data. All data data used to generate the figures are included in the manuscript and supporting files; source data files have been provided for *Figures 2–6* and their supplements.

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
