## [Editor Report · eLife assessment]

The work by Hornberger and team presents a novel workflow for the visualisation of myofibrils with high resolution and contrast that will be highly valued by the scientific community. The methods include **solid** validation of both sample preparation and analysis, and have been used to make the **fundamental** discovery of myofibrillogenesis as the mechanism of mechanical loading-induced growth. Whether this mechanism is present in other settings of muscle growth (i.e., non-loading), other striated tissue (e.g myocardium), or is sex-dependent, will require future experiments.

---

## [Referee Report · Reviewer #1 (Public Review)]

Summary:

Using a state-of-the-art image analysis pipeline the authors report that muscle cell hypertrophy in mice and humans occurs primarily through an increase in the number of myofibrils (myofibrillogenesis) and not myofibril hypertrophy.

Strengths:

A strength of the study is the development and validation of an automated image analysis pipeline to quantify myofibril size and abundance in mouse and human muscle cells. In addition to the pipeline, which requires relatively readily available microscopy equipment (an additional strength) is the development of a methodology to optimally prepare muscle samples for high-resolution imaging.

Weaknesses:

A weakness of the study was that only one time-point was assessed during hypertrophy. As mentioned by the authors, this precluded an assessment of the myofibril splitting mechanism.

---

## [Referee Report · Reviewer #2 (Public Review)]

Summary:

In this work, the authors sought to (1) establish a method for measuring muscle fiber subcellular structure (myofibrils) using common, non-specialized laboratory techniques and equipment, and (2) use this method to provide evidence on whether loading-induced muscle fiber growth was the result of myofibril growth (of existing myofibrils) or myofbrillogenesis (creation of new myofibrils) in mice and humans. The latter is a fundamental question in the muscle field. The authors succeeded in their aims and provided useful methods for the muscle field and detailed insight into muscle fiber hypertrophy; specifically, that loading-induced muscle fiber hypertrophy may be driven mostly by myofibrillogenesis.

Strengths:

1. The usage of murine and human samples to provide evidence on myofibril hypertrophy vs myofibrillogenesis.

2. A nice historical perspective on myofibrillogenesis in skeletal muscle.

3. The description of a useful and tractable IHC imaging method for the muscle biology field supported by extensive validation against electron microscopy.

4. Fundamental information on how myofiber hypertrophy ensues.

Weaknesses:

- The usage of young growing mice (8-10 weeks) versus adult mice (>4 months) in the murine mechanical overload experiments, as well as no consideration for biological sex. The former point is partly curtailed by the adult human data that is provided (male only). Still, the usage of adult mice would be preferable for these experiments given that maturational growth may somehow affect the outcomes. For the latter point, it is not clear whether male or female mice were used.

---

## [Referee Report · Reviewer #3 (Public Review)]

Summary:

Radial muscle growth involves an increase in overall muscle cross-sectional area. For decades this process has been described as the splitting of myofibrils to produce more myofibrils during the growth process. However, a closer look at the original papers shows that the evidence underlying this description was incomplete. In this paper, the authors have developed a novel method using fluorescence microscopy to directly measure myofibril size and number. Using a mouse model of mechanical loading and a human model of resistance exercise they discovered that myofibrillogenesis is playing a key role in the radial growth of muscle fibers.

Strengths:

1. Well-written and clear description of hypothesis, background, and experiments.

2. Compelling series of experiments.

3. Different approaches to test the hypothesis.

4. Rigorous study design.

5. Clear interpretation of results.

6. Novel findings that will be beneficial to the muscle biology field.

7. Innovative microscopy methods that should be widely available for use in other muscle biology labs.

---

## [Author Response]

The following is the authors’ response to the original reviews.

**Reviewer #2 (Public Review)**
Weaknesses1. The usage of young growing mice (8-10 weeks) versus adult mice (>4 months) in the murine mechanical overload experiments. The usage of adult mice would be preferable for these experiments given that maturational growth may somehow affect the outcomes.

The basis for this critique is not clear as it has been shown that the longitudinal growth of bones is complete by ⁓8 weeks of age (e.g., PMID: 28326349, and 31997656). These studies, along with others, also indicate that 8 weeks is a post-pubescent age in mice. For these reasons, 8 weeks of age was viewed as being representative of the human equivalent of when people start to perform resistance exercise with the goal of increasing muscle mass. Also, it’s important to consider that the mice were 10-12 weeks of age when the muscles were collected which would be equivalent to a human in their lower 20’s. In our human study, the mean age of the subjects was 23. Given the above points, it’s hard for us to appreciate why the use of mice that started at 8-10 weeks of age is viewed as a weakness. With that being said, we recognize that there may be age-related changes in mechanisms of mechanical load-induced growth, but it was not our intent to address this topic.

(1b) No consideration for biological sex.

We appreciate this point and we agree that sex is an important variable to consider. In this study, we explored an unchartered topic and therefore we wanted to minimize as many known variables as possible. We did that, in part, by focusing specifically on male subjects. In the future, it will certainly be important to explore whether sex (and age) impact the structural adaptations that drive the mechanical load-induced growth of muscle fibers.

1. Information on whether myofibrillogenesis is dependent on hypertrophy induced by loading, or just hypertrophy in general. To provide information on this, the authors could use, for instance, inducible Myostatin KO mice (a model where hypertrophy and force production are not always in lockstep) to see whether hypertrophy independent from load induces the same result as muscle loading regarding myofibrillogenesis.

This is a great suggestion, but it goes beyond the intended scope of our study. Nevertheless, with the publication of our FIM-ID methodology, the answer to this and related questions can now be obtained in a time- and cost-effective manner.

1. Limited information on Type 1 fiber hypertrophy. A "dual overload" model is used for the mouse where the soleus is also overloaded, but presumably, the soleus was too damaged to analyze. Exploring hypertrophy of murine Type 1 fibers using a different model (weight pulling, weighted wheel running, or forced treadmill running) would be a welcome addition.

The point is well taken and further studies that are aimed at determining whether there are differences in how Type I vs. Type II fibers grow would be an excellent subject for future studies.

**Reviewer #3 (Public Review)**
1. Supplemental Figure 1 is not very clear.

Supplemental Figure 1 is now presented as Supplemental Figure 2. We carefully reexamined this figure and, in our opinion, the key points have been appropriately conveyed. We would be more than happy to revise the figure, but we would need guidance with respect to which aspect(s) of the figure were not clear to the reviewer.

**Reviewer #1 (Recommendations For The Authors)**
Introduction.1. I do not think the first paragraph is really necessary. Cell growth is a fundamental property of cell biology that requires no further justification.

We believe that it is essential to remind all readers about the importance of skeletal muscle research. For some, the detrimental impact of skeletal muscle loss on one’s quality of life and the greater burden on the healthcare system may not be known.

1. I prefer "fundamental" over "foundationally".

All mentions of the word “foundational” and “foundationally” have been changed to “fundamental” and “fundamentally.”

1. As usual for the Hornberger lab, the authors do an excellent job of providing the (historical) context of the research question.

Thank you for this positive comment.

1. I prefer “Goldspink” as “Dr. Goldspink” feels too personal especially when you are critical of his studies.

All instances of “Dr.” have been removed when referring to the works of others. This includes Dr. Goldspink and Dr. Tokuyasu.

1. Fourth paragraph, after reference #17. I felt like this discussion was not necessary and did not really add any value to the introduction.

We believe that this discussion should remain since it highlights the widely accepted notion that mechanical loading leads to an increase in the number of myofibrils per fiber, yet there is no compelling data to support this notion. This discussion highlights the need for documented evidence for the increase in myofibril number in response to mechanical loading and, as such, it serves as a major part of the premise for the experiments that were conducted in our manuscript.

1. The authors do a nice job of laying out the challenge of rigorously testing the Goldspink model of myofiber hypertrophy.

Thank you!

Results1). For the EM images, can the authors provide a representative image of myofibril tracing? From the EM image provided, it is difficult to evaluate how accurate the tracing is.

-Representative images and an explanation of myofibril calculation have been provided in Supplemental Figure 5.

1. In the mouse, how does the mean myofibril CSA compare between EM and FIM-ID?

**Author response image 1. sa4fig1:** 

The above figures compare the myofibril CSA and fiber CSA measurements that were obtained with EM and FIM-ID for all analyzed fibers, as well as the same fibers separated according to the fiber type (i.e., Ox vs. Gly). The above figure shows that the FIM-ID measurements of myofibril CSA were slightly, yet significantly, lower than the measurements obtained with EM. However, we believe that it would be misleading to present the data in this manner. Specifically, as shown in Fig. 4C, a positive linear relationship exists between myofibril CSA and fiber CSA. Thus, a direct comparison of myofibril CSA measurements obtained from EM and FIM-ID would only be meaningful if the mean CSA of the fibers that were analyzed were the same. As shown on the panel on the right, the mean CSA of the fibers analyzed with FIM-ID was slightly, yet significantly, lower than the mean CSA of the fibers analyzed with EM. As such, we believe that the most appropriate way to compare the measurements of the two methods is to express the values for the myofibril CSA relative to the fiber CSA and this is how we presented the data in the main figure (i.e., Fig. 4E).

1. Looking at Fig. 3D, how is intermyofibrillar space calculated when a significant proportion of the ROI is odd-shaped myofibrils that are not outlined? It is not clear how the intermyofibrillar space between the odd-shaped myofibrils is included in the total intermyofibrillar space calculation for the fiber.

The area occupied by the intermyofibrillar components is calculated by using our custom “Intermyofibrillar Area” pipeline within CellProfiler. Briefly, the program creates a binary image of the SERCA signal. The area occupied by the white pixels in the binary image is then used to calculate the area that is occupied by the intermyofibrillar components. To help readers, an example of this process is now provided in supplemental figure 4.

1. What is the average percentage of each ROI that was not counted by CP (because a myofibril did not fit the shape criteria)? The concern is that the method of collection is biasing the data. In looking at EM images of myofibrils (from other studies), it is apparent that myofibrils are not always oval; in fact, it appears that often myofibrils have a more rectangular shape. These odd-shaped myofibrils are excluded from the analysis yet they might provide important information; maybe these odd-shaped myofibrils always hypertrophy such that their inclusion might change the overall conclusion of the study. I completely understand the challenges of trying to quantify odd-shaped myofibrils. I think it is important the authors discuss this important limitation of the study.

First, we would like to clarify that myofibrils of a generally rectangular shape were not excluded. The intent of the filtering steps was to exclude objects that exhibited odd shapes because of an incomplete closure of the signal from SERCA. To illustrate this point we have annotated the images from Figure 3B-D with a red arrow which points to a rectangular object and blue arrows which point to objects that most likely consisted of two or more individual myofibrils that were falsely identified as a single object.

**Author response image 2. sa4fig2:** 

We appreciate the reviewer's concern that differences in the exclusion rates between groups could have biased the outcomes. Indeed, this was something that we were keeping a careful eye on during our analyses, and we hope that the reviewer will take comfort in knowing that objects were excluded at a very similar rate in both the mouse and human samples (44% vs. 46% for SHAM vs. MOV in mice, and 47% vs. 47% for PRE vs. POST in humans). We realize that this important data should have been included in our original submission and it is now contained with the results section of the revised version of our manuscript. Hopefully the explanation above, along with the inclusion of this data, will alleviate the reviewers concerns that differences between the groups may have been biased by the filtering steps.

Discussion.1. I think the authors provided a balanced interpretation of the data by acknowledging the limitation of having only one time-point. i.e., not being able to assess the myofibril splitting mechanism.

Thank you!

1. I think a discussion on the important limitation of only quantifying oval-shaped myofibrils should be included in the discussion.

Please refer to our response to comment #4 of the results section.

**Reviewer #2 (Recommendations For The Authors)**
Overall, this is a thoughtful, clear, and impactful manuscript that provides valuable tools and information for the skeletal muscle field. My specific comments are as follows:1. In the introduction, I really appreciate the historical aspect provided on myofbrillogenesis. As written, however, I was expecting the authors to tackle the myofibril "splitting" question in greater detail with their experiments given the amount of real estate given to that topic, but this was not the case. Consider toning this down a bit as I think it sets a false expectation.

We acknowledge that the study does not directly address the question about myofibril splitting. However, we believe that it is important to highlight the background of this untested theory since it serves as a major part of the premise for the experiments that were performed.

1. In the introduction, is it worth worth citing this study? https://rupress.org/jcb/articlepdf/111/5/1885/1464125/1885.pdf.

This is a very interesting study but, despite the title, we do not believe that it is accurate to say that this study investigated myofibrillogenesis. Instead (as illustrated by the author in Fig. 9) the study focused on the in-series addition of new sarcomeres at the ends of the pre-existing myofibrils (i.e., it studied in-series sarcomerogenesis). In our opinion, the study does not provide any direct evidence of myofibrillogenesis, and we are not aware of any studies that have shown that the chronic stretch model employed by the authors induces myofibrillogenesis. However, numerous studies have shown that chronic stretch leads to the in-series addition of new sarcomeres.

1. Is there evidence for myofbrillogenesis during cardiac hypertrophy that could be referenced here?

This is a great question, and one would think that it would have been widely investigated. However, direct evidence for myofibrillogenesis during load-induced cardiac hypertrophy is just as sparse as the evidence for myofibrillogenesis during load-induced skeletal muscle hypertrophy.

1. In the introduction, perhaps mention that prolonged fixation is another disadvantage of EM tissue preparation. This typically prevents the usage of antibodies afterwards, whereas the authors have been able to overcome this using their method, which is a great strength.

Thank you for the suggestion. This point has been added the 5th paragraph of the introduction.

1. In the introduction, are there not EM-compatible computer programs that could sidestep the manual tracing and increase throughput? Why could software such as this not be used? https://www.nature.com/articles/s41592-019-0396-9

While we agree that automated pipelines have been developed for EM, such methods require a high degree of contrast between the measured objects. With EM, the high degree of contrast required for automated quantification is rarely observed between the myofibrils and the intermyofibrillar components (especially in glycolytic fibers). Moreover, one of the primary goals of our study was to develop a time and cost-effective method for identifying and quantifying myofibrils. As such, we developed a method that would not require the use of EM. We only incorporated EM imaging and analysis to validate the FIM-ID method. Therefore, utilizing an EM-compatible program to sidestep the manual tracing would have sped up the validation step, but it would not have accomplished one of the primary goals of our study.

1. In the results, specifically for the human specimens, were "hybrid" fibers detected and, if so, how did the pattern of SERCA look? Also, did the authors happen to notice centrallynucleated muscle fibers in the murine plantaris after overload? If so, how did the myofibrils look? Could be interesting.

For the analysis of the human fibers, two distinct immunolabeling methods were performed. One set of sections was stained for SERCA1 and dystrophin, while the other set was stained for SERCA2 and dystrophin. In other words, we did not perform dual immunolabeling for SERCA1 and SERCA2 on the same sections. Therefore, during the analysis of the human fibers, we did not detect the presence of hybrid fibers. Furthermore, while we did not perform nuclear staining on these sections, it should be noted that nuclei do not contain SERCA, and to the best of our recollection, we did not detect any SERCAnull objects within the center of the fibers. Moreover, our previous work has shown that the model of MOV used in this study does not lead to signs of degeneration/regeneration (You, Jae-Sung et al. (2019). doi:10.1096/fj.201801653RR). Therefore, it can be safely assumed that very few (if any) of the fibers analyzed in this study were centrally nucleated.

1. In the Results, fixed for how long? This is important since, at least in my experience, with 24+ hours of fixation, antibody reactivity is significantly reduced unless an antigen retrieval step is performed (even then, not always successful). Also, presumably these tissues were drop-fixed? These details are in the Methods but some additional detail here could be warranted for the benefit of the discerning and interested reader.

For both the mouse and human, the samples were immersion-fixed (presumably the equivalent of “drop-fixed”) in 4% paraformaldehyde in 0.1M phosphate buffer solution for a total of 24 hours (as described in the Methods section). We agree that prolonged aldehyde fixation can affect antibody reactivity; however, the antibodies used for FIM-ID did not require an antigen retrieval step.

1. In the results regarding NADH/FAD autofluorescence imaging, a complimentary approach in muscle was recently described and could be cited here:https://journals.physiology.org/doi/full/10.1152/japplphysiol.00662.2022.

We appreciate the reviewer’s recommendation to add this citation for the support of our method for fiber type classification and have added it to the manuscript in the second paragraph under the “Further refinement and validation of the automated measurements with FIM-ID” subsection of the Results as citation number 57.

1. In the results, "Moreover, no significant differences in the mean number of myofibrils per fiber CSA were found when the results from the FIM-ID and EM-based measurements were directly compared, and this point was true when the data from all analyzed fibers was considered..." Nit-picky, but should it be "were considered" since data is plural?

Thanks, this error was corrected.

1. In the discussion, are the authors developing a "methodology" or a "method"? I think it may be the latter.

We agree that “method” is the correct term to use. Instances of the word “methodology” have been replaced with “method.”

1. In the discussion, since the same fibers were not being tracked over time, I'm not sure that saying "radial growth" is strictly correct. It is intuitive that the fibers were growing during loading, of course, but it may be safer to say "larger fibers versus control or the Pre sample" or something of the like. For example, "all the fiber types that were larger after loading versus controls" as opposed to "showed significant radial growth"

While we agree that the fiber size was not tracked over time, the experiments were designed to test for a main effect of mechanical loading. Therefore, we are attributing the morphological adaptations to the mechanical loading variable (i.e., mechanical loadinduced growth). The use of terms like “the induction of radial growth” or “the induction of hypertrophy” are commonly used in studies with the methods employed in this study. Respectfully, we believe that it would be more confusing for the readers if we used the suggested terms like "all the fiber types that were larger after loading versus controls". For instance, if I were the reader I would think to myself… but there fiber types that were larger than others before loading (e.g., Ox vs. Gly), so what are the authors really trying to talk about?

1. I would suggest making a cartoon summary figure to complement and summarize the Methods/Results/Discussion

Thank you for this suggestion. We created a cartoon that summarizes the overall workflow for FIM-ID and this cartoon is now presented in Supplemental Figure 1.